# Dynamical Mechanical and Thermal Analyses of Biodegradable Raw Materials for Additive Manufacturing

**DOI:** 10.3390/ma13081819

**Published:** 2020-04-12

**Authors:** Simona-Nicoleta Mazurchevici, Andrei-Danut Mazurchevici, Dumitru Nedelcu

**Affiliations:** 1Department of Machine Manufacturing Technology, “Gheorghe Asachi” Technical University of Iasi, Blvd. Mangeron, No. 59A, 700050 Iasi, Romania; simona0nikoleta@yahoo.com (S.-N.M.); andrei0maz@yahoo.com (A.-D.M.); 2Academy of Romanian Scientists, Str. Ilfov, Nr. 3, Sector 5, 010164 Bucharest, Romania

**Keywords:** 3D printing, biodegradable materials, DMA, DSC, SEM, EDAX, XRD

## Abstract

In order to find new ways to ensure sustainable development on a global level, it is essential to combine current top technologies, such as additive manufacturing, with the economic, ecological, and social fields. One objective of this paper refers to wire manufacture such as Arboblend V2 Nature, Arbofill Fichte, and Arboblend V2 Nature reinforced with Extrudr BDP “Pearl” (BDP—Biodegradable Plastic) in order to replace the plastic materials. After wire manufacture by extrusion, the diameter accuracy was analyzed compared with the Fiber Wood wire using SEM analyses and also EDAX—Energy Dispersive X-ray Analysis and DSC—Differential Scanning Calorimetry analyses were done in order to identify their elemental composition and the phase transitions suffered by the materials during heating. Using the samples obtained through the Fused Deposition Modeling (FDM) method, both crystalline phases and chemical composition information (XRD analysis) were identified, as well was determined the visco-elastic behavior Dynamic Mechanical Analysis (DMA), for the reinforced material and Fiber Wood. The extruded wires have allowed size for the printing equipment, around 1.75 mm with tolerance of ± 0.05 mm. The wire material diagrams, Arboblend V2 Nature reinforced with Extrudr BDP “Pearl” and Fiber Wood following the calorimetric analysis, presented peaks corresponding to material crystallization, while Arbofill Fichte revealed only the melting temperature. The storage module was almost double in case of Arboblend V2 Nature reinforced with Extrudr BDP “Pearl” compared with Fiber Wood and materials’ melting temperatures were confirmed by the analyses carried out.

## 1. Introduction

Today, additive manufacturing technologies are considered revolutionary manufacturing technology with high growth potential as well as high performance manufacturing. According to the additive manufacturing (AM) definition, these manufacturing processes involve the three-dimensional parts’ manufacture through successively thin layers’ addition until the part is complete. According to refs. [1,2], not only the AM technologies are being widely used for the manufacture of nonfunctional/functional parts by using a variety of materials such as metals, polymers, ceramics, and also combinations of these ones. The refs. [1,2,3,4,5,6,7,8,9,10] turn out that the prototyped parts could find their application in various fields such as automotive, engineering, industrial design, aerospace, architecture, construction, military, medical and dental industries, biotech (human tissue replacement), and many other fields.

Due to the excessive use of products made of petroleum based-plastic materials, the development of a bio-based material becomes a necessity for many industrial applications. As the authors [11,12,13] also state, the design of bio-based materials should minimize the environmental pollution but also to compete with the functional characteristics of the parts made of synthetic polymers. The main naturally occurring polymers, part of carbohydrate family, are starch and cellulose (polysaccharide). Natural fibers are used as reinforcements in composites. Among the most used fibers are as hemp, flax, straw, jute, kenaf and lignin, cellulose, and hemicellulose [11,12,13].

The most important starting point for the production of parts from biodegradable materials is gaining a better understanding of their structure and various properties, and identifying plastic material parts that can be successfully substituted by biodegradable materials. In this regard, ref. [14] gives us an overview of all the aspects mentioned above. Thus, taking into account all the studies presented by ref. [14], regarding the properties or behavior of biopolymers, such as polylactic acids (PLA), starch-based materials, and biocomposites based on lignin, cellulose, and natural plant fiber, the idea of replacing plastics with biodegradable materials was developed, without any recommendations for the materials proposed to be studied in this paper.

Given the constant need to ensure sustainable development on the global level, the idea of merging top manufacturing technologies, such as 3D printing with the ecological, economic, and social fields, has emerged. Also, ref. [15] raises the issue of relatively reduced components’ service life caused by the varied customers’ demands and, also, the frequent changes in what concerns the product design. This study covered a wide range of fused deposition modeling applications and advancements by using standard materials (as acrylonitrile butadiene styrene, polylactic acids, polyamides, and others), advanced materials (4D materials), and application-specific materials (composite feedstock filaments), but it does not mention at all the lignin matrix biodegradable polymers.

Innovation in the field of biodegradable materials leads to a reduction in the use of materials based on fossil resources, which are so damaging to the environment. These aspects were pointed out by ref. [16], making an overview of PLA, the most used biodegradable material in Fused Deposition Modeling (FDM), but there is no reference to plastics based on renewable raw materials/cellulose material.

The Fused Deposition Modeling (FDM) method is the additive manufacturing technology that uses, as raw material, the wire produced by the extrusion process. The extrusion and prototyping of biodegradable material use the same equipment as for conventional plastics, as pointed out [17,18]. Another benefit provided by this category is the reduction in energy consumption due to lower prototyping temperatures compared with the common plastics. This last advantage was also reported by ref. [19], who analyzed from mechanically, structurally, and morphologically points of view the Arboblend V2 Nature material injected samples. Arboblend V2 Nature is a lignin-based material. The injection molding temperature set during the process was low, indeed, between 150–170 °C compared to many synthetic plastics’ temperatures. The correlation of the results regarding the thermal behavior obtained by the above-mentioned researchers with the results obtained in this paper on the DSC (Differential Scanning Calorimetry) analysis of the Arboblend V2 Nature raw material, in the form of granules, made it possible to establish the conditions much faster in order to prototype this material.

The materials analyzed in this research are all included in the category of eco-friendly materials which can replace different plastics. Thus, Arboblend V2 Nature materials are designed to be biodegradable or resistant, depending on the intended application, and Arbofill Fichte materials are high-quality compounds made from renewable raw materials and plastics and both with lignin matrix as a major constituent apart from the different annual plant fibers (flax, hemp, sisal, etc.) and natural additives according to the information provided by the producer [20].

Arboblend V2 Nature reinforced with Extrudr BDP “Pearl” (BDP—Biodegradable Plastic) is a biocomposite made by the authors of this paper in order to increase 3D printing processability and technological properties. The reinforcement, i.e., Extrudr BDP “Pearl” is also a biodegradable material, from renewable resources, easy to print, and with good mechanical properties (tensile strength). Fiber Wood filament is a thermoplastic material, made entirely from natural wood, which is part of its constituent elements, being produced by Fiberlogy. Fiber Wood is PLA with added wood fibers and, therefore, has many of the same properties as PLA, the most used biodegradable material in the FDM printing process.

The paper’s aim consisted of wire manufacture by extrusion of Arboblend V2 Nature reinforced with Extrudr BDP “Pearl”, the diameter accuracy measurement compared with wire Fiber Wood diameter through SEM analysis. Also, in order to identify the composition and phase transitions of the studied materials during the heating process, we used the Energy Dispersive X-ray Analyze (EDAX) and DSC analyze. Using the FDM technology, we obtained printed parts from Arboblend V2 Nature reinforced with Extrudr BDP “Pearl” as samples that will be studied and compared with printed Fiber Wood samples. The novelty of the paper is highlighted by the new composite material as a mixture between Arboblend V2 Nature (as a base material) and Extrudr BDP “Pearl” (as reinforcement material), wire manufactured from base material and reinforcement material and the results such as: The wire thermal behavior of Arboblend V2 Nature base material and reinforced with Extrudr BDP “Pearl”, the wire thermal behavior of Arbofill Fichte, the dynamic mechanical behavior of Arboblend V2 Nature reinforced with Extrudr BDP “Pearl”.

The significance of this work consists of the possibility to replace parts from plastic materials, taking into account the negative influence of these on the environment.

In fact, Arboblend V2 Nature has the possibility of replacing synthetic plastics, from a tensile strength point of view [19]. These can substitute PE—40 MPa, PVDF (poly vinylidene fluoride)—43 MPa, and PCTFE (ethylene copolymer)—32 MPa. The Arboblend V2 Nature tensile strength mean value is 44.05 ± 0.48 MPa. Regarding the friction coefficient, 0.1376 for the biodegradable material can substitute with success the PA 12 (polyamide)—(0.32–0.38), PP (polypropylene)—0.3, PE-HD (high density polyethylene)—0.29, ABS (acrylonitrile butadiene styrene)—0.5, and PVDF (poly vinylidene fluoride)—0.23.

## 2. Materials and Methods

Four biodegradable materials were used for this research, namely:-Wires of Arboblend V2 Nature, Arbofill Fichte, Arboblend V2 Nature reinforced with Extrudr BDP “Pearl” (80% with 20% in weight), obtained from extrusion of granules (produced by the company Tecnaro, (Ilsfeld, Germany) in the laboratory of Fine Mechanics and Nanotechnologies, “Gheorghe Asachi” Technical University of Iasi, Romania, using Noztek Touch equipment (produced by Noztek, Great Britain, Shoreham-by-Sea).-Wire of Fiber Wood, purchased and produced by Fiberlogy (Brzezie, Polonia).

The parameters set on the extrusion equipment in order to obtain the 1.75 mm wire diameter size are presented in Table 1.

Furthermore, in order to obtain the specific samples for Dynamic Mechanical Analysis (DMA), the Raise 3D Pro2 Plus 3D equipment produced by Raise3D, USA, was used to obtain the samples with dimensions of (25 × 4 × 2) mm, which were standardized by the manufacturer of the DMA 242 Artemis NETZSCH machine (NETZSCH producer, Selb, Germany).

The 3D printing was done using the Fused Deposition Modeling (FDM) technology. The nozzle diameter was 0.4 mm and the printing wire diameter was 1.75 mm. All the samples were printed with 100% infill. Table 2 presents the temperatures of the extrusion nozzle and of the printing table for both materials.

The morphological analysis of the extruded wires was performed using the SEM and EDAX methods on a QUANTA 200 3D electron microscope produced by FEI Company, Fremont, CA, USA.

The XRD analysis, for studying the crystalline structure, was used to identify the material crystalline phases and, thus, the information regarding chemical composition was revealed. The phases were identified by comparing the data obtained with those from the database references.

For the DSC analysis, we used up to 5-mm-long sample fragments cut from all sample types, weighing less than 50 mg. For this purpose, a NETZSCH differential scanning calorimeter type DSC 200 F3 Maya (NETZSCH producer, Selb, Germany), was used, with sensitivity of <l W, temperature accuracy of 0.1 K, and enthalpy accuracy, generally <1%. The device was calibrated with Bi, In, Sn, and Zn standards. Temperature scans were performed between 20–200 °C with a heating rate of 10 K/min, under A_r_ protective atmosphere. The DSC thermograms recorded during heating were evaluated with Proteus software, provided by NETZSCH, using tangent method. The temperatures were determined where half of the transformation took place (T50) as well as the amount of dissipated/absorbed heat, respectively.

The Dynamic Mechanical Analysis (DMA) was conducted utilizing DMA 242 Artemis NETZSCH equipment with a three-point bending specimen holder to characterize flexural behavior by temperature scans and strain sweeps. The temperature scans were performed between room temperature (RT) and 373.15 K (100 °C) from three to three Kelvin, with a 5 N dynamic force, a deformation of 50 μm and a frequency of 1 Hz. The DMA diagrams, recorded during temperature scans, display the variations of storage modulus (E’) and internal friction (determined as the ratio between loss and storage modulus, tan d = E”/E’), during a heating cycle.

## 3. Results and Discussions

### 3.1. The SEM Analysis

The background SEM images for all the materials studied were on a 100X and the details were on a 500X magnification scale.

It can be observed that all the three wires obtained allowed for dimensions to be printed by a 3D prototyping equipment, with the nozzle diameter of 1.75 mm and a tolerance of ± 0.05 mm. Thus, Arboblend V2 Nature obtained a wire size of 1.74 mm, Arbofill Fichte was 1.72 mm, and Arboblend V2 Nature reinforced with Extrudr BDP “Pearl” was a 1.76 mm diameter, Figure 1a–c. The SEM analysis for the Arbofill Fichte material, Figure 1b, revealed its structure. Therefore, in the background image as well as in detail, numerous (natural) fibers embedded in a polymer can be noticed (according to the granule manufacturer’s specifications, Tecnaro), this being a lignin-based biocomposite. The diameters’ values of the above materials were compared with the diameter value of Fiber Wood filament (Figure 1d).

### 3.2. The EDAX Chemical Analysis

Figure 2 presents the electron microscopy images and the chemical characterization of the extruded wires as well as of the fiber wood wire.

The below graphs show the results obtained from energy dispersive X-ray analysis (EDAX) regarding the mass and atomic percentage of the chemical elements of wires from Arboblend V2 Nature (Figure 2a), Arbofill Fichte (Figure 2b), Arboblend V2 Nature reinforced with Extrudr BDP “Pearl” (Figure 2c), and Fiber Wood (Figure 2d) granules.

According to the obtained results from this analysis that identifies the material chemical elements, we distinguished as dominant constituents the carbon and the oxygen (in atomic and mass percentage) elements which, in the case of three of the biodegradable materials studied, Arboblend V2 Nature, Arboblend V2 Nature reinforced with Extrudr BDP “Pearl”, and Fiber Wood, were found in approximately equal proportions. Their presence in such a large quantity confirms their high biodegradation rate because they contain chemical elements that are abundantly found in the chemical structure of plants, under different types of oxygen-carbon bonds (specific to cellulose, hemicellulose, lignin, lignin derivatives, etc.). Arbofill Fichte material revealed a mass and atomic ratio of approximately 82% carbon and 18% oxygen, which was most likely due to the presence of natural vegetable fibers in its structure.

### 3.3. The XRD Analysis

The X-ray diffraction analysis is the most commonly used technique to characterize the crystallinity and the phase purity of a material. Thus, the X-ray powder diffraction analysis was performed for the Arboblend V2 Nature reinforced with Extrudr BDP “Pearl” and for Fiber Wood materials, the prototyped samples. For the sample made from Arboblend V2 Nature reinforced with Extrudr BDP “Pearl” (Figure 3), the results of the specific technical literature were used in order to find the chemical compound that was identified as major peak, at 17 2θ angles.

Thus, according to studies conducted by Gupta [21], regarding the preparation and characterization of lignin nanofibers by electrospinning technique during the X-ray analysis, two major peaks associated with semicrystalline lignin were obtained, at 2θ = 17° and, respectively, at 2θ = 19°. Also, the presence of this peak could be attributed to the vegetable natural fibers present in the material composition.

For the Fiber Wood sample (Figure 4), the major peak was identified at an angle of 36.93 2θ with intensity over 10,000 for carbon, a chemical element that can be abundantly found in the composition of natural fibers.

The XRD diffraction showed a maximum peak in the angular range 2θ angle = 15–25, for the Arboblend V2 Nature reinforced with Extrudr BDP “Pearl” material, and the angular range 2θ angle = 11–44 for the Fiber Wood material, clearly attesting to the existence of the crystalline phase in the examined biodegradable thermoplastic materials. The identification of peaks had not yet been possible; however, based on the data obtained from the specific technical literature as well as from its thermal analysis, the above conclusion regarding the occurrence of the crystalline phase was confirmed.

From the graphs’ analysis, the presence of crystallized substances was observed. This aspect was in full agreement with the study of materials’ thermal behavior, revealing that these have a semicrystalline structure.

### 3.4. The DSC Analysis

In order to determine the heating behavior of the biodegradable materials studied (the transitions), the wire samples were heated with 10 K/min up to 200 °C, and the thermal analysis was completed before it started to deteriorate. The melting temperature of materials was determined for this analysis in order to establish the optimum 3D printing temperatures.

In Figure 5, it was realized an overlapping of the graphics for the Arboblend V2 Nature and Arboblend V2 Nature reinforced with Extrudr BDP “Pearl” materials in order to highlight the influence of the reinforcement on the DSC curves appearance.

For Arboblend V2 Nature, the DSC curve shows two endothermic transformations in the analyzed temperature range, 40–190 °C. The first endothermic transformation recorded at 58.1 °C could be associated to the melting temperature of the polycaprolactone (PCL) component [22] to the melting temperature of natural wax [23] or to the gelatinization of the starch when it is heated in presence of water [24]. We mention that all these components are found in the composition of this material according to the details provided by those who have patented this material [3], Tecnaro, a biopolymer company, and Fraunhofer Institute for Chemical Technology. At the 172.8 °C temperature, an endothermic peak corresponding to the material melting point was identified. The heat amount absorbed for the first transformation was −0.69 kJ/kg and for the second it was −49.52 kJ/kg.

The thermal analysis of the Arboblend V2 Nature reinforced with Extrudr BDP “Pearl” wire highlighted at 62.2 °C temperature an endothermic peak was associated with the melting or gelatinization of some Arboblend V2 Nature constituent components [20,22,23,24]. This transition was also revealed by the DSC curve of the Arboblend V2 Nature wire, but with a slight displacement. The heat amount absorbed in this case was −2.94 kJ/kg. At a 149.4 °C temperature, a minimal endotherm appeared associated with the Extrudr BDP “Pearl” melting point. This assertion is based on further DSC analyses performed by the authors of this paper, which reported the melting temperature of the reinforcement at 156 °C. The heat amount absorbed for this transformation was −0.79 kJ/kg. The last minimum endotherm located at 172 °C was assigned to the melting of the Arboblend V2 Nature core material and it had an absorbed heat of −29.93 kJ/kg. This temperature was similar to the previous analysis of the Arboblend V2 Nature material.

Reinforcement of Arboblend V2 Nature with Extrudr BDP “Pearl” led to displacement of the first transformation by 4.1 °C and an increase in the absorbed heat amount from −0.69 kJ/kg to −2.94 kJ/kg, the appearance of an additional transformation at 149.4 °C temperature, and decreased approximately by half of the absorbed heat amount needed to melt the material Arboblend V2 Nature.

Figure 6 shows the DSC thermogram recorded during the controlled heating of the Arbofill Fichte material. In this case, within the analyzed temperature range, 40–190 °C, there was only one minimum endotherm corresponding to the melting point, at 148.5 °C temperature. The amount of heat absorbed in this case was −25.7 kJ/kg.

The calorimetric analysis of the Fiber Wood material is presented in the thermogram of Figure 7. In this case, within the analyzed temperature range, the DSC thermogram had two endothermic minima and an exothermic minimum, behavior similar to that of Arboform^®^ LV3 Nature “liquid wood” [25].

The Fiber Wood wire (Figure 7) at the temperature of 61.1 °C revealed a minimal caloric energy absorption peak, which corresponded to the residual liquid medium removal. The amount of heat absorbed was −8.38 kJ/kg. The exothermic peak, highlighted at 97.9 °C, can be associated with the lignin reticular rearrangement (the base Fiber Wood material compound/the matrix), the process taking place with an amount of heat released of 18.76 kJ/kg. The melting point of the material was recorded at 153.7 °C, and it can also be noticed that there was also a small endothermic peak in the 138–175 °C temperature range that corresponded to the melting point of another component, i.e., natural fibers, from the composition of the analyzed material, or due to the Fiber Wood material premelting (fractional melting). The minimum low endotherm took place absorbing a heat quantity of −0.55 kJ/kg, respectively, −2.94 for the one corresponding to the temperature of 153.7 °C.

Regarding the behavior of the studied materials, compared to other synthetic plastics used in 3D printing, the biodegradable materials, according to the DSC analysis, had a lower melting point than the synthetic ones, thus leading to the reduction of energy consumption. The materials that can be substituted from the thermal point of view are: ABS (220–250 °C), flexible (225–245 °C), HIPS (high-impact polystyrene) (230–245 °C), naylon (220–270 °C), PC (polycarbonate) (260–310 °C), PP (polypropylene) (220–250 °C), metal-filled (190–220 °C), and other [26].

### 3.5. The DMA Analyses

The Dynamic Mechanical Analysis (DMA) determined the elastic modulus, also named storage modulus, G’, the viscous modulus named loss modulus, G”, and the damping coefficient, tan δ, as a function of temperature, frequency, or time.

Figure 8 and Figure 9 illustrate, through DMA thermograms, the variations of storage modulus and internal friction during heating.

The effects of a similar chemical composition of raw materials are quite evident if we compare the thermograms of the samples analyzed, which had a rather similar behavior.

Both samples had a solid-state transition during heating, because the internal friction (tan δ) had two maximum peaks (internal friction), one for each material, at temperatures of 334.2 K/61.05 °C (for Arboblend V2 Nature reinforced with Extrudr BDP “Pearl”) and 336.8 K/63.65 °C (for Fiber Wood), while the storage module (E’) reached a maximum value of 952 MPa for Fiber Wood and 1542 MPa for the reinforced material. Both biodegradable materials showed quite similar phenomena characterized by a maximum of E’ at heating temperatures around 60 °C.

It is also worth mentioning that the maximum peaks recorded by the both materials subjected to DMA overlapped at the endothermic peaks, corresponding to the glass transitions of the material, revealed by their thermal analysis. It should be mentioned that, in the case of some calorimetric analyses, the vitrification temperature could not be identified because the glass transition of the lignin, which was difficult to determine due to its structure and broad heterogeneity as well as to its molecular weight. According to other researchers [5], following the thermal behavior analysis of the biodegradable material, namely, Arboform LV3 Nature, it was reported that Tg, in the case of different types of lignin, can vary between 100–180 °C, depending on the plant species of and on the lignin extraction procedure from these.

From a DMA point of view, the Arboblend V2 Nature reinforced with Extrudr BDP “Pearl” can substitute materials as Tango Black plus and DM9895 with values of storage modulus between 1300–1800 MPa [27].

## 4. Conclusions

Due to humanity’s current need for a faster replacement of plastic materials, a new direction of research was developed using biodegradable materials which, combined with the latest technologies, can offer comparable or even better products than those made from plastics. Thus, in this paper, various analyses were carried out with the above mentioned purpose.

The assessment of the SEM images obtained in the cases of all four types of biodegradable materials, Arboblend V2 Nature, Arbofill Fichte, Arboblend V2 Nature reinforced with Extrudr BDP “Pearl”, and Fiber Wood, confirmed that the diameters of the wires obtained by extrusion were within the limits allowed by the printing equipment, 1.75 mm and tolerance of ±0.05 mm.

Following the chemical analysis, three of the biodegradable materials studied were shown to have a similar chemical composition, if we consider the amount of oxygen and carbon contained. The biodegradable material Arbofill Fichte presented a much higher percentage of carbon, due to its composition having a higher quantity of natural fibers.

The diffractogram analysis obtained by XRD analysis revealed that both materials (Arboblend V2 Nature reinforced with Extrudr BDP “Pearl” and Fiber Wood) had a semicrystalline structure.

The wire material diagrams, Arboblend V2 Nature reinforced with Extrudr BDP “Pearl” and Fiber Wood following the calorimetric analysis, presented peaks corresponding to material crystallization, while Arbofill Fichte revealed only the melting temperature. The data regarding the material melting points were used in order to set the technological parameter “printing temperature” correctly, for a better adhesion of the layers successively deposited in order to obtain samples with technological properties comparable to those of conventional plastics.

Comparing the results regarding the chemical structure of materials used for the injected samples [19] and for the prototyped ones, it was concluded that the printing process parameters did not change significantly the thermal behavior and structure of the base material Arboblend V2 Nature and of the reinforced material Arboblend V2 Nature with Extrudr BDP “Pearl”.

Following the determination of the visco-elastic behavior in the case of samples made from the reinforced material and from Fiber Wood, the storage module was found to reach a maximum value of 1542 MPa for Arboblend V2 Nature reinforced with Extrudr BDP “Pearl” and a smaller value, 952 MPa, for the Fiber Wood biodegradable material. Moreover, the material crystallization phase was confirmed by comparing the results obtained from DSC and DMA analyses.

## Figures and Tables

**Figure 1 materials-13-01819-f001:**
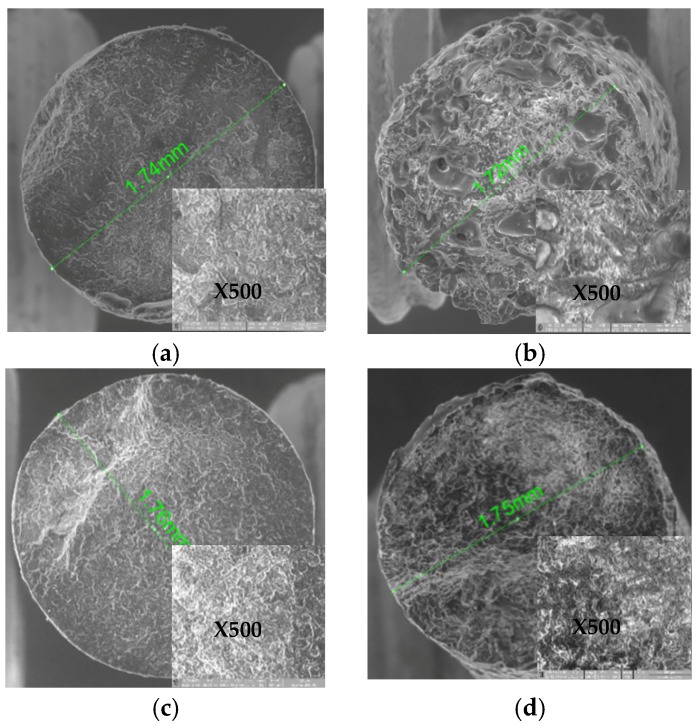
The SEM wires images: (**a**) Arboblend V2 Nature, (**b**) Arbofill Fichte, (**c**) Arboblend V2 Nature reinforced with Extrudr BDP “Pearl”, (**d**) Fiber Wood.

**Figure 2 materials-13-01819-f002:**
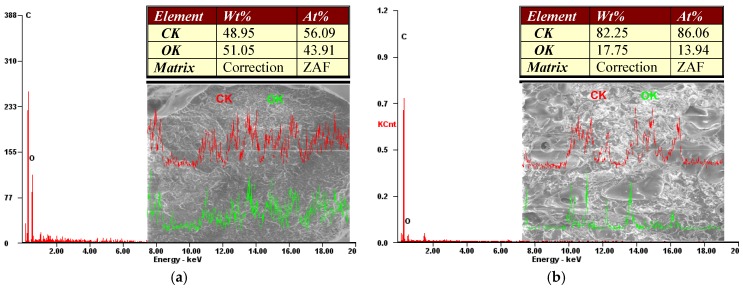
The EDAX (Energy Dispersive X-ray Analysis) wire images: (**a**) Arboblend V2 Nature, (**b**) Arbofill Fichte, (**c**) Arboblend V2 Nature reinforced with Extrudr BDP “Pearl”, (**d**) Fiber Wood.

**Figure 3 materials-13-01819-f003:**
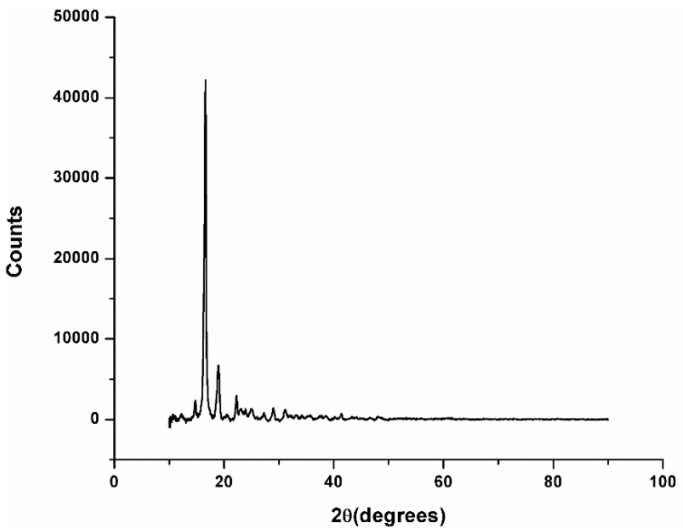
XRD analyses for Arboblend V2 Nature reinforced with Extrudr BDP “Pearl” material.

**Figure 4 materials-13-01819-f004:**
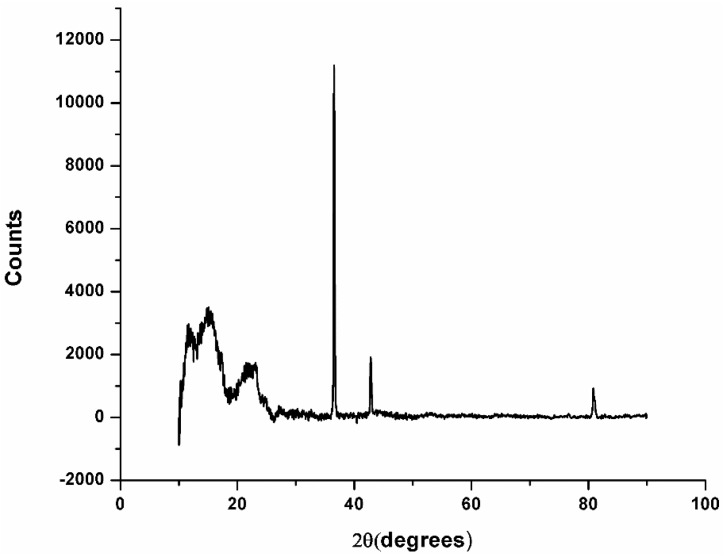
XRD analyses for Fiber Wood material.

**Figure 5 materials-13-01819-f005:**
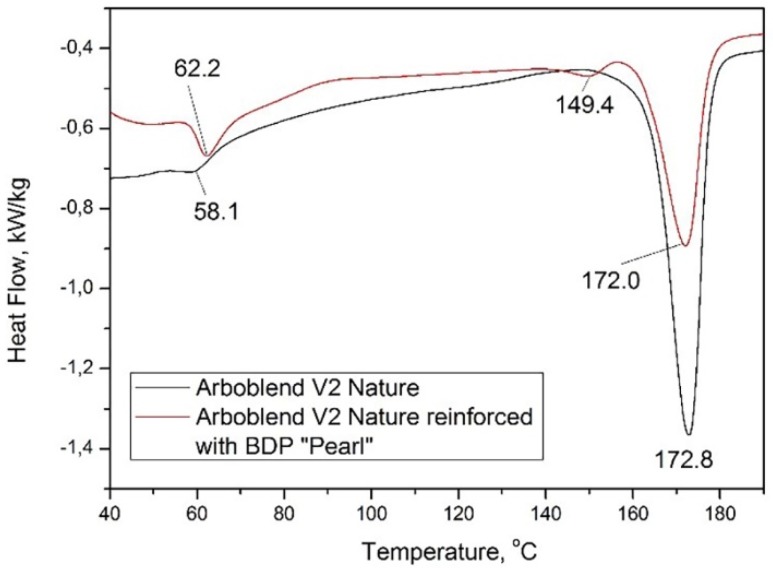
Highlighting the main thermal behavior of the wires made from Arboblend V2 Nature and Arboblend V2 Nature reinforced with Extrudr BDP “Pearl”.

**Figure 6 materials-13-01819-f006:**
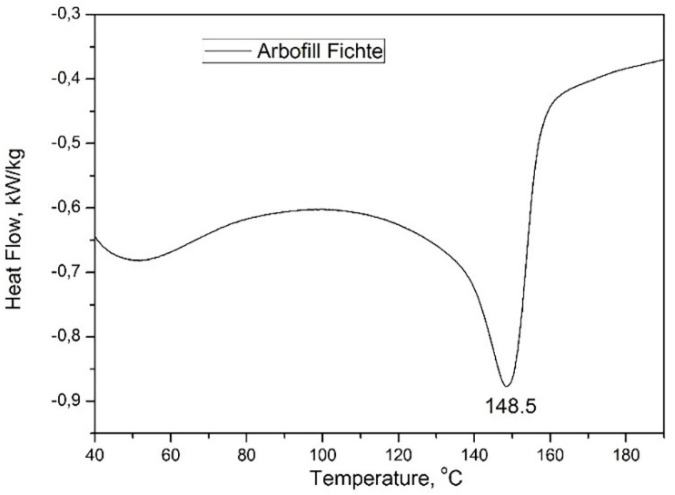
Highlighting the main thermal behavior of the wire made from Arbofill Fichte.

**Figure 7 materials-13-01819-f007:**
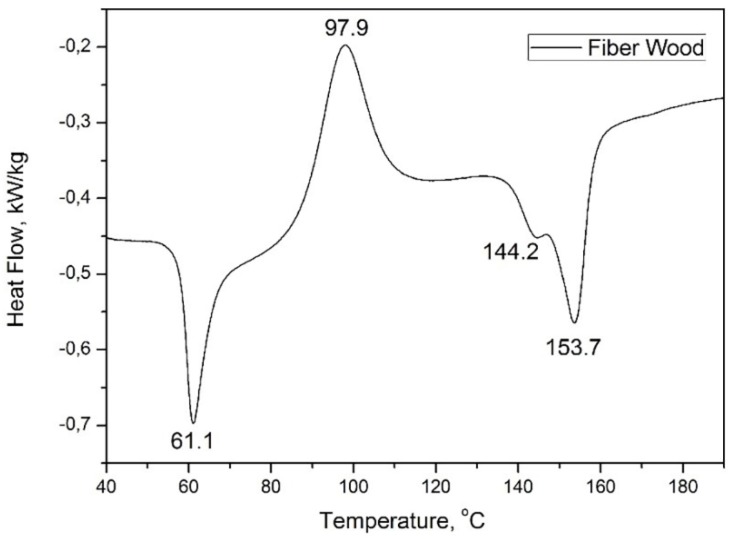
Highlighting the main thermal behavior of the Fiber Wood wire.

**Figure 8 materials-13-01819-f008:**
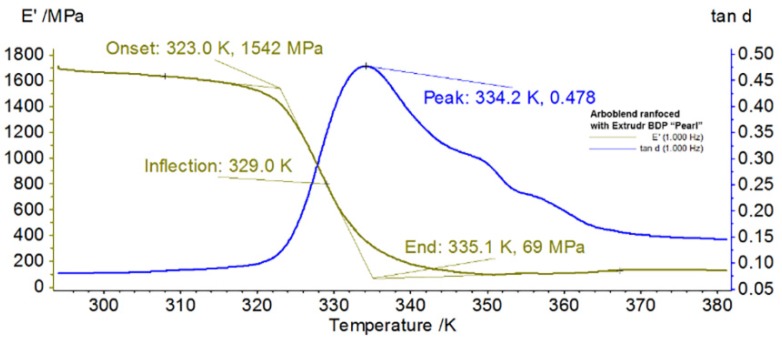
Dynamic Mechanical Analysis (DMA) thermogram recorded during heating of Arboblend V2 Nature reinforced with Extrudr BDP “Pearl”.

**Figure 9 materials-13-01819-f009:**
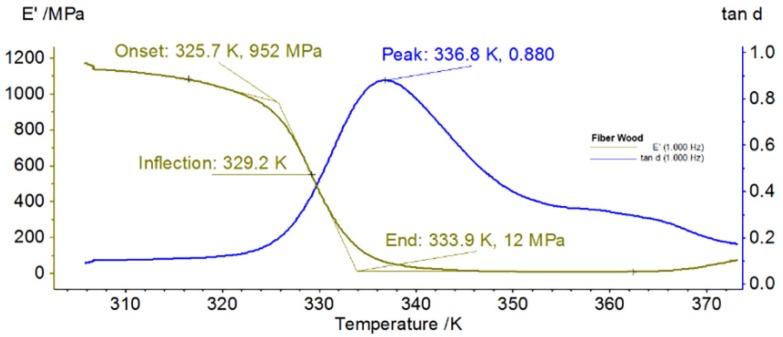
DMA thermogram recorded during fiber wood heating.

**Table 1 materials-13-01819-t001:** The parameters’ values used for biodegradable material extrusion.

	Process Parameters		
Extruded Materials	1st Temperature (°C)	2nd Temperature (°C)	The Advance Speed (rot/min)
Arboblend V2 Nature	150	155	10
Arbofill Fichte	155	160	10
Arboblend V2 Nature reinforced with Extrudr BDP “Pearl”	155	160	15

**Table 2 materials-13-01819-t002:** The temperatures of extrusion nozzle and printing table, for the two prototyped materials.

No.	Material	Nozzle Temperature (°C)	Printing Table Temperature (°C)
1	Arboblend V2 Nature reinforced with BDP “Pearl”	205	65
2	Fiber Wood	200	55

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
