# Peer review of "Dynamical Mechanical and Thermal Analyses of Biodegradable Raw Materials for Additive Manufacturing"

_materials, 2020, doi:10.3390/ma13081819_

Round 1

Reviewer 1 Report

The manuscript concerns very interesting analyzes regarding the replacement of plastics with biodegradable plastics. Mechanical and thermal analyzes of biodegradable raw materials for additive manufacturing were carried out.

The title of the manuscript corresponds to the presented content.

The abstract of the manuscript contains the necessary information.

Introduction could be expanded. In its current form is not enough.

Materials and methods require little correction. Not everything is clearly outlined.

The results are presented legibly. A small note regarding the resolution of some charts.

The discussion should be extensive. Miss few references to other research related to the introduction of biodegradable plastics.

In addition to the summary of the research, I suggest to present the most important conclusions resulting from the research.

References should be better chosen. The literature presented is not current. No reference to the latest research results.

Author Response

Response to Reviewer 1

Point 1: Introduction could be expanded. In its current form is not enough.

Response 1: The introduction chapter was improved by adding new references and informations.

Lines 29-42: Today, additive manufacturing technologies are considered revolutionary manufacturing technology. According to the additive manufacturing (AM) definition, these manufacturing processes involve the three-dimensional parts manufacture through successively thin layers' addition until the part is complete. According to the [1, 2], and not only, the AM technologies are being widely used for the manufacture of non-functional / functional parts by using a variety of materials such as metals, polymers, ceramics and also combinations of these ones. The [1, 2, 3, 4, 5, 6, 7, 8, 9,10] turn out that the prototyped parts could find their application in various fields such as automotive, engineering, industrial design, aerospace, architecture, construction, military, medical and dental industries, biotech (human tissue replacement) and many other fields.

Due to the excessive use of products made of petroleum based-plastic materials the development of a bio-based material become a necessity. As the authors [11, 12, 13] also state, the design of bio-based materials should minimize the environmental pollution but also to compete with the functional characteristics of the parts made of synthetic polymers. The main natural occurring polymers, part of carbohydrate family are starch and cellulose (polysaccharide). Natural fiber are used as reinforcements in composites, among the most used fibers are as hemp, flax, straw, jute, kenaf and lignin, cellulose and hemicellulose, [11, 12, 13].

Lines 52-61: Also, [15] raise the issue of relatively reduced components service life caused by the varied customers’ demands and, also, the frequent changes in what concerns the product design. This study covers a wide range of fused deposition modeling applications and advancements by using standard materials (as acrylonitrile butadiene styrene, poly lactic acids, polyamides, and others) advanced materials (4D materials), application-specific materials (composite feedstock filaments) but this one, it does not mention at all the lignin matrix biodegradable polymers.

Innovation in the field of biodegradable materials leads to a reduction in the use of materials based on fossil resources, which are so damaging to the environment. These aspects were pointed by [16], making an overview of PLA, the most used biodegradable material in FDM printing, but there is no reference to plastics based on renewable raw materials/ cellulose material.

Lines 97 – 103: “As for the possibility of replacing synthetic plastics with the Arboblend V2 Nature material, from tensile strength point of view, [19], these one can replace PE-40 MPa, PVDF (poly vinylidene fluoride)-43 MPa and PCTFE (ethylene copolymer)-32 MPa. The Arboblend V2 Nature tensile strength mean value is 44.05 ± 0.48 MPa, the material had a brittle behavior. Regarding the friction coefficient, 0.1376 for the biodegradable material, can substitute with success the PA 12 (polyamide)-(0.32-0.38), PP (polypropylene) - 0.3, PE-HD (high density polyethylene)-0.29, ABS (Acrylonitrile butadiene styrene) - 0.5 and PVDF (poly vinylidene fluoride)-0.23.”

Point 2: Materials and methods require little correction. Not everything is clearly outlined.

Response 2: The “Materials and methods” chapter was improved by adding additional information about equipment’s and not only.

Lines 107-Line 110: “Wires of Arboblend V2 Nature, Arbofill Fichte, Arboblend V2 Nature reinforced with Extrudr BDP "Pearl" (80% with 20% in weight). obtained from extrusion of granules (produced by the German company Tecnaro) in the laboratory of Fine Mechanics and Nanotechnologies, “Gheorghe Asachi” Technical University of Iasi, Romania, using Noztek Touch equipment (produced by Noztek, Great Britain).”

Lines 111-112: “Wire of Fiber Wood, purchased and produced by Fiberlogy.

The parameters set on the extrusion equipment in order to obtain the”

Line 118: “produced by Raise3D, USA”

Lines 119-120: “(NETZSCH producer, Selb, Germany)”

Line 128: “produced by FEI Company, USA”

Line 134: “(NETZSCH producer, Selb, Germany)”

Point 3: The results are presented legibly. A small note regarding the resolution of some charts.

Response 3: Was improved the resolution (300dpi) of the next figures: Figure 1; Figure 2; Figure 5; Figure 6; Figure 7; Figure 8; Figure 9

Point 4: The discussion should be extensive. Miss few references to other research related to the introduction of biodegradable plastics.

Response 4: Was introduced 3 new references related to biodegradable materials, reference [11], [12] and [13].

Lines 37-42: “Due to the excessive use of products made of petroleum based-plastic materials the development of a bio-based material become a necessity. As the authors [11, 12, 13] also state, the design of bio-based materials should minimize the environmental pollution but also to compete with the functional characteristics of the parts made of synthetic polymers. The main natural occurring polymers, part of carbohydrate family are starch and cellulose (polysaccharide). Natural fiber are used as reinforcements in composites, among the most used fibers are as hemp, flax, straw, jute, kenaf and lignin, cellulose and hemicellulose, [11, 12, 13]."

Point 5: In addition to the summary of the research, I suggest to present the most important conclusions resulting from the research.

Response 5: The abstract has been improved.

Lines 21-23: “The wire material diagrams, Arboblend V2 Nature reinforced with Extrudr BDP "Pearl" and Fiber Wood following the calorimetric analysis, presented peaks corresponding to material crystallization, while Arbofill Fichte revealed only the melting temperature.”

Point 6: References should be better chosen. The literature presented is not current. No reference to the latest research results.

Response 6: Was added 19 new references from the last 8 years.

Lines 331-360:

  1. Shahrubudina, N., Leea, T.C., Ramlana, R. An Overview on 3D Printing Technology: Technological, Materials, and Applications. In Procedia Manufacturing, 2nd International Conference on Sustainable Materials Processing and Manufacturing (SMPM 2019), 2019, 35, 1286–1296.
  2. Rupinder Singh, Gurchetan Singh, Jaskaran Singh, Ranvijay Kumar. 3D Printing of Polyether-Ether-Ketone Functional Prototypes for Engineering Applications Encyclopedia of Renewable and Sustainable Materials 2020, 5, 207-214.
  3. Hassan, M., Dave, K., Chandrawati, R., Dehghani, F., Gomes, V. G. 3D printing of biopolymer nanocomposites for tissue engineering: Nanomaterials, processing and structure-function relation. European Polymer Journal 2019, 121, 109340.
  4. Won Jun Choi, Ki Seob Hwang, Hyuk Jun Kwon, Chanmin Lee, Jun-Young Lee. Rapid development of dual porous poly(lactic acid) foam using fused deposition modeling (FDM) 3D printing for medical scaffold application. Materials Science and Engineering: C 2020, 110, 110693.
  5. Ngo, T. D., Kashani, A., Imbalzano, G., Nguyen, K. T. Q., Hui, D. Additive manufacturing (3D printing): A review of materials, methods, applications and challenges. Composites Part B: Engineering 2018, 143, 172-196.
  6. Joshi, S. C., Abdullah Sheikh, A. 3D printing in aerospace and its long-term sustainability. 2015 Virtual and Physical Prototyping, 10(4), 1-11.
  7. Ballard, D. H, Trace, A. P., Ali, S., Hodgdon, T., Zygmont M. E., DeBenedectis, C. M., Smith, S. E., Richardson, M. L., Patel, M. J., Decker, S. J., Lenchik L. Clinical Applications of 3D Printing: Primer for Radiologists. Academic Radiology. 2018, 25(1), 52-65.
  8. US Army developing process for using 3D printing at depots and in the field. Available online: https://www.defensenews.com/land/2020/02/04/us-army-developing-process-for-using-3d-printing-at-depots-and-in-the-field/ (accessed on 15 March 2020).
  9. Busra Tugce Duymaz, Fatma Betul Erdiler, Tugba Alan, Mehmet Onur Aydogdu. 3D bio-printing of levan/polycaprolactone/gelatin blends for bone tissue engineering: Characterization of the cellular behavior. European Polymer Journal 2019, 119, 426-437.
  10. Praveen Vasamsetty, Tejaswini Pss, Divya Kukkala, Madhavi Singamshetty, Shashivardhan Gajula. 3D printing in dentistry – Exploring the new horizons. Materials Today: Proceedings 2020 (in press).
  11. Ebnesajjad, S. Handbook of Biopolymers and Biodegradable Plastics: Properties, Processing, and Applications, Publisher: Elsevier, William Andrew, 2012, pp.109-128.
  12. Greene, J. P. Sustainable Plastics: Environmental Assessments of Biobased, Biodegradable and Recycled Plastics, Publisher: Hoboken, New Jersey: Wiley, 2014.
  13. Ashter, S. A. Introduction to Bioplastics Engineering, 1st ed., Publisher: William Andrew, 2016, pp. 19 – 30.

Lines 363-370:

  1. Singh, R., Davim, J. P. Additive Manufacturing: Applications and Innovations, Publisher: CRC Press Taylor & Francis Group, 2018.
  2. Davim, J. P. Additive and Subtractive Manufacturing: Emergent Technologies, Publisher: Walter de Gruyter GmbH & Co KG, Aveiro, 2020, vol.4, pp. 59-112.
  3. Redwood, B., Schoffer, F., Garret, B. The 3D Printing Handbook Technologies, design and applications, Ed. Hardcover, Amsterdam, The Netherlands, 2017.
  4. Carausu, C, Mazurchevici, A., Ciofu, C., Mazurchevici, S. The 3D printing modelling of biodegradable material. IOP Conf. Series: Materials Science and Engineering 2018, 400, 042008.

Lines 384-387:

  1. Filament Properties Table. Available online: https://www.simplify3d.com/support/materials-guide/properties-table/ (accessed on 12 october 2019).
  2. Zhang, B., Kowsari, K., Serjouei, A., Dunn, M.L., Ge, Q. Reprocessable thermosets for sustainable threedimensional printing. Nature Communications 2018, DOI: 10.1038/s41467-018-04292-8.

All the changes were highlighted in the paper, with gray color.

Reviewer 2 Report

In this manuscript, the authors characterized the mechanical and thermal performance of 4 types of biodegradable materials based on SEM, XRD, DSC, DMA, and other techniques. Although the manuscript presents a variety of work aiming to underpin the advantages of liquid wood materials compared to conventional plastic counterparts, the authors did not conclude the study through the comparison within different liquid wood materials and with plastics. Therefore, the authors should address the abovementioned issue along with several other concerns before the consideration for publication.

  • The introduction is difficult to follow and does not contain essential arguments to support the importance of using liquid wood materials. Additionally, the references are a little scarce, more references should be included. Why only four types of materials are studied, what is so special in them?
  • Moreover, the authors should improve the English in the manuscript, especially in the introduction section. Some sentences are very difficult to understand.
  • How to prepare Arboblend V2 Nature reinforced materials?
  • Please explain the results of EDAX chemical analysis in section 3.2 and XRD analysis in section 3.3.
  • Through the DSC analysis described in 3.4 and DMA analyses in 3.5, are the performance of these biodegradable materials competitive compared to plastics?

Author Response

Response to Reviewer 2

Point 1: In this manuscript, the authors characterized the mechanical and thermal performance of 4 types of biodegradable materials based on SEM, XRD, DSC, DMA, and other techniques. Although the manuscript presents a variety of work aiming to underpin the advantages of liquid wood materials compared to conventional plastic counterparts, the authors did not conclude the study through the comparison within different liquid wood materials and with plastics. Therefore, the authors should address the abovementioned issue along with several other concerns before the consideration for publication.

Response 1: Was added a short comment concerning the comparison between liquid wood and plastics on the end of the introduction chapter.

Lines 97 – 103: “As for the possibility of replacing synthetic plastics with the Arboblend V2 Nature material, from tensile strength point of view, [19], these one can replace PE-40 MPa, PVDF (poly vinylidene fluoride)-43 MPa and PCTFE (ethylene copolymer)-32 MPa. The Arboblend V2 Nature tensile strength mean value is 44.05 ± 0.48 MPa, the material had a brittle behavior. Regarding the friction coefficient, 0.1376 for the biodegradable material, can substitute with success the PA 12 (polyamide)-(0.32-0.38), PP (polypropylene) - 0.3, PE-HD (high density polyethylene)-0.29, ABS (Acrylonitrile butadiene styrene) - 0.5 and PVDF (poly vinylidene fluoride)-0.23.”

Point 2: The introduction is difficult to follow and does not contain essential arguments to support the importance of using liquid wood materials. Additionally, the references are a little scarce, more references should be included. Why only four types of materials are studied, what is so special in them?

Response 2: We used the materials from this paper because:

-Extrudr BDP Pearl as reinforcement because this material has mechanically and thermally comparable to PLA (https://www.extrudr.com/en/products/catalogue/pearl-natur_2277/).

-Arboblend V2 Nature was a base material because it has higher biodegradation rate compared with Arbofill Fichte;

-Fiber Wood is PLA with added wood fibers and therefore has many of the same properties as PLA, the most used biodegradable material in FDM printing process.

Point 3: Moreover, the authors should improve the English in the manuscript, especially in the introduction section. Some sentences are very difficult to understand.

Response 3: Was The English was improved by a specialized company.

Point 4: How to prepare Arboblend V2 Nature reinforced materials?

Response 4: The reinforced material was obtained in weight with 20% Extrudr BDP Pearl and 80% Arboblend V2 Nature.

Lines 107-Line 110: “Wires of Arboblend V2 Nature, Arbofill Fichte, Arboblend V2 Nature reinforced with Extrudr BDP "Pearl" (80% with 20% in weight). obtained from extrusion of granules (produced by the German company Tecnaro) in the laboratory of Fine Mechanics and Nanotechnologies, “Gheorghe Asachi” Technical University of Iasi, Romania, using Noztek Touch equipment;”

Point 5: Please explain the results of EDAX chemical analysis in section 3.2 and XRD analysis in section 3.3.

Response 5: We made some additional comments from this point of view.

Lines 172-180: “According to the obtained results from this analyse that identify the material chemical elements, have been distinguished as dominant constituents the carbon and the oxygen (in atomic and mass percentage) elements which, in the case of three of the biodegradable materials studied Arboblend V2 Nature, Arboblend V2 Nature reinforced with Extrudr BDP "Pearl" and Fiber Wood are found in approximately equal proportions. Their presence in such a large quantity confirms their high biodegradation rate because contain chemical elements that are abundantly found in the chemical structure of plants, under different types of oxygen-carbon bonds (specific to cellulose, hemicellulose, lignin, lignin derivatives, and not only). Arbofill Fichte material reveals a mass and atomic ratio of approximately 82% carbon and 18% oxygen, which is most likely due to the presence of natural vegetable fibers in its structure.”

Point 6: Through the DSC analysis described in 3.4 and DMA analyses in 3.5, are the performance of these biodegradable materials competitive compared to plastics?

Response 6: Was added a comment regarding the comparison between the used materials and other polymers.

Lines 260-264: Regarding the behavior of the studied materials, compared to other synthetic plastics used in 3D printing, the biodegradable materials, according to the DSC analysis, have a lower melting point than the synthetic ones, thus leading to the reduction of energy consumption. The materials that can be substituted from the thermal point of view are: ABS (220-250⁰C), Flexible (225-245⁰C), HIPS - High Impact Polystyrene (230-245⁰C), Naylon (220-270⁰C), PC- polycarbonate (260-310⁰C), PP – polypropylene (220-250⁰C), Metal Filled (190-220⁰C), and other, [26].

Lines 290-291: From DMA point of view, the Arboblend V2 Nature reinforced with Extrudr BDP "Pearl" can substitute materials as Tango Black plus and DM9895 with values of storage modulus between (1300-1800)MPa, [27].

All the changes were highlighted in the paper, with gray color.

Reviewer 3 Report

The manuscript from the title: "Dynamical mechanical and thermal analyses of biodegradable raw materials for additive manufacturing" shows a good effort in the characterisation process underlined by the use of many techniques over the chosen materials. However,  all the materials tested are, to the extent of my understanding, industrially produced polymers and, probably with the exceptions of the reinforced one, basic characterisation is normally already available by the producing company and the knowledge of the dimension of the filament after extrusion is more of practical that of scientific interest. English level is good even though the reading is not always easy, check must be done on frequent typos and sentences are sometimes written in such a way that comprehension becomes difficult. The references are very few and not very up to date, the most recent is of 2016.

Overall I think the research in order to be published needs more polymers to be taken into account with a deep investigation on the difference that the presence of more than one reinforcement charge (ore more than a shape or distribution into the matrix) would make.

In any case, I will give some other comments below in order to a possible improvement of the paper:

Line 33 - Him, who? Please always refer to an article by the name of the first author plus et al. in case they are more than one author (e.g. Pilla s. ) or the number in the reference list. 

Line 33/37 - Please rephrase this sentence is not clear what you are stating.

Line 46 - No need to specify that Nedelcu e Comaneci are the authors, please delete the word 

Line 47 - What is the Arboblend V2 Nature? Even though you have the reference it should be briefly explained what it is. Is it a commercial product, biodegradable etc.

p.s. I see that you describe the material from line 53 to 63, please give a brief explanation anyways at this point, the reader otherwise has to comprehend it from the beginning. It will be sufficient to say "Arboblend V2 Nature, a lignin based material,"

Line 47 - What do you mean with injected samples? Are the sample injected with the polymer or you meant extruded? Please clarify.

Line 48 - Many plastics is referred to synthetic plastics, isn't is so? Please specify.

Line 56 - In the introduction I would rather refer to an anonymous "producer" instead of the Tecnaro German company, which in any case should be have a more complete reference, and specify the company name, product name etc in the material an methods.

Line 58-63 - Is not clear what Fiber wood is, at the beginning I thought it was a constituent of the Extrude BDP "Pearl", but reading the next sentence is cleat that is a polymer per se that you are comparing with the Arboblend V2 Nature. Thus, please rephrase explaining what Fiber wood, meaning why you are using it, why you use it as a comparison, is it the market choice for biodegradable extrusion polymers?  

Line 71-71 - Can you please explain do you mean with wire manufacture from based material and reinforced materials?

Line 73 - Why you didn't reinforce also the Arbofill Fichte liquid wood? If it's just a commercial product what is the reason to study its thermal behaviour, it should be sell with a technical sheet. And in any case what would be the novelty in this?

Line 75 - You should explain better which synthetic plastic materials are you willing to substitute and what you mean with "parts from plastic materials" otherwise is a to broad concept.

Line 75-76 - "This is a research target for the authors of this article in the next period. " is a sentence that will be more appropriate at the end of the conclusion after you already showed the novelty reached by the research.

Line 80 - please add a were in front of obtained.

Line 82 - What is a Noztek touch equipment? who  produce it and where? Please specify.

LIne 84 - 80% with 20% have no meaning if written in this form, please specify if it's % in volume or in weight and in case add a respectively in order to exactly state what the % refers to.

Line 85 - Why don't you put the explanation of the equipment used and producer before the list so you do not have to repeat it? Please change.

Line 87 - The parameters chosen are not technological, but settings parameters of the extruder, I would rather refer to them in this way or, considering are just two parameters I would just list them.

Line 92-94 - Please specify what are the equipment you are referring too, what they do and from which company are produced and where. 

Line 103 - Please specify from which company are produced and where. 

Line 109 - Please specify from which company are produced and where. 

Line 115 - Please specify from which company are produced and where.

Paragraph "The SEM Analysis":

You lack of giving any discussion on the figure 1d and also the other 3 figures are not deeply discussed. Is not very clear in this way what is the point in your study of having SEM micrography.

Paragraph "The EDAX Chemical Analysis":

Is quite not clear why you are doing an EDAX and the X-ray on the samples as you already know the basic chemical composition and you are not doing any process that could change it, moreover the two techniques give the same result with the X-ray being normally more precise of the EDAX. 

The only difference could be the line scan analysis that you performed with the EDAX, but being the elements presents all composed of C and O the difference is very small between reinforced and not reinforced samples. Also the technique is not precise enough to make some speculations over the position of the fibres reinforcement when the same elements are present.

Author Response

Response to Reviewer 3

Point 1: The manuscript from the title: "Dynamical mechanical and thermal analyses of biodegradable raw materials for additive manufacturing" shows a good effort in the characterisation process underlined by the use of many techniques over the chosen materials. However, all the materials tested are, to the extent of my understanding, industrially produced polymers and, probably with the exceptions of the reinforced one, basic characterisation is normally already available by the producing company and the knowledge of the dimension of the filament after extrusion is more of practical that of scientific interest. English level is good even though the reading is not always easy, check must be done on frequent typos and sentences are sometimes written in such a way that comprehension becomes difficult. The references are very few and not very up to date, the most recent is of 2016.

Response 1: Was added 19 new references from the last 8 years.

  1. Shahrubudina, N., Leea, T.C., Ramlana, R. An Overview on 3D Printing Technology: Technological, Materials, and Applications. In Procedia Manufacturing, 2nd International Conference on Sustainable Materials Processing and Manufacturing (SMPM 2019), 2019, 35, 1286–1296.
  2. Rupinder Singh, Gurchetan Singh, Jaskaran Singh, Ranvijay Kumar. 3D Printing of Polyether-Ether-Ketone Functional Prototypes for Engineering Applications Encyclopedia of Renewable and Sustainable Materials 2020, 5, 207-214.
  3. Hassan, M., Dave, K., Chandrawati, R., Dehghani, F., Gomes, V. G. 3D printing of biopolymer nanocomposites for tissue engineering: Nanomaterials, processing and structure-function relation. European Polymer Journal 2019, 121, 109340.
  4. Won Jun Choi, Ki Seob Hwang, Hyuk Jun Kwon, Chanmin Lee, Jun-Young Lee. Rapid development of dual porous poly(lactic acid) foam using fused deposition modeling (FDM) 3D printing for medical scaffold application. Materials Science and Engineering: C 2020, 110, 110693.
  5. Ngo, T. D., Kashani, A., Imbalzano, G., Nguyen, K. T. Q., Hui, D. Additive manufacturing (3D printing): A review of materials, methods, applications and challenges. Composites Part B: Engineering 2018, 143, 172-196.
  6. Joshi, S. C., Abdullah Sheikh, A. 3D printing in aerospace and its long-term sustainability. 2015 Virtual and Physical Prototyping, 10(4), 1-11.
  7. Ballard, D. H, Trace, A. P., Ali, S., Hodgdon, T., Zygmont M. E., DeBenedectis, C. M., Smith, S. E., Richardson, M. L., Patel, M. J., Decker, S. J., Lenchik L. Clinical Applications of 3D Printing: Primer for Radiologists. Academic Radiology. 2018, 25(1), 52-65.
  8. US Army developing process for using 3D printing at depots and in the field. Available online: https://www.defensenews.com/land/2020/02/04/us-army-developing-process-for-using-3d-printing-at-depots-and-in-the-field/ (accessed on 15 March 2020).
  9. Busra Tugce Duymaz, Fatma Betul Erdiler, Tugba Alan, Mehmet Onur Aydogdu. 3D bio-printing of levan/polycaprolactone/gelatin blends for bone tissue engineering: Characterization of the cellular behavior. European Polymer Journal 2019, 119, 426-437.
  10. Praveen Vasamsetty, Tejaswini Pss, Divya Kukkala, Madhavi Singamshetty, Shashivardhan Gajula. 3D printing in dentistry – Exploring the new horizons. Materials Today: Proceedings 2020 (in press).
  11. Ebnesajjad, S. Handbook of Biopolymers and Biodegradable Plastics: Properties, Processing, and Applications, Publisher: Elsevier, William Andrew, 2012, pp.109-128.
  12. Greene, J. P. Sustainable Plastics: Environmental Assessments of Biobased, Biodegradable and Recycled Plastics, Publisher: Hoboken, New Jersey: Wiley, 2014.
  13. Ashter, S. A. Introduction to Bioplastics Engineering, 1st ed., Publisher: William Andrew, 2016, pp. 19 – 30.

Lines 363-370:

  1. Singh, R., Davim, J. P. Additive Manufacturing: Applications and Innovations, Publisher: CRC Press Taylor & Francis Group, 2018.
  2. Davim, J. P. Additive and Subtractive Manufacturing: Emergent Technologies, Publisher: Walter de Gruyter GmbH & Co KG, Aveiro, 2020, vol.4, pp. 59-112.
  3. Redwood, B., Schoffer, F., Garret, B. The 3D Printing Handbook Technologies, design and applications, Ed. Hardcover, Amsterdam, The Netherlands, 2017.
  4. Carausu, C, Mazurchevici, A., Ciofu, C., Mazurchevici, S. The 3D printing modelling of biodegradable material. IOP Conf. Series: Materials Science and Engineering 2018, 400, 042008.

Lines 384-387:

  1. Filament Properties Table. Available online: https://www.simplify3d.com/support/materials-guide/properties-table/ (accessed on 12 october 2019).
  2. Zhang, B., Kowsari, K., Serjouei, A., Dunn, M.L., Ge, Q. Reprocessable thermosets for sustainable threedimensional printing. Nature Communications 2018, DOI: 10.1038/s41467-018-04292-8.

Point 2: Overall I think the research in order to be published needs more polymers to be taken into account with a deep investigation on the difference that the presence of more than one reinforcement charge (ore more than a shape or distribution into the matrix) would make.

Response 2: The authors were focus only on these materials because:

-Extrudr BDP Pearl as reinforcement because this material has mechanically and thermally comparable to PLA (https://www.extrudr.com/en/products/catalogue/pearl-natur_2277/). 

-Arboblend V2 Nature was a base material because it has higher biodegradation rate compared with Arbofill Fichte;

-Fiber Wood is PLA with added wood fibers and therefore has many of the same properties as PLA, the most used biodegradable material in FDM printing process.

Point 3: In any case, I will give some other comments below in order to a possible improvement of the paper:

Line 33 - Him, who? Please always refer to an article by the name of the first author plus et al. in case they are more than one author (e.g. Pilla s. ) or the number in the reference list.

Response 3: Was changed in [1] instead of him.

Point 4: Line 33/37 - Please rephrase this sentence is not clear what you are stating.

Response 4: Yes, the lines 46-49 was rephrase.

Lines 46-49: “Thus, taking into account all the studies presented by [14], regarding the properties or behavior of biopolymers, such as Poly Lactic Acids (PLA), starch based materials, and bio composites based on lignin, cellulose, natural plant fiber, the idea of replacing plastics with biodegradable materials has been developed, without any recommendations for the materials proposed to be studied in this paper.”

Point 5: Line 46 - No need to specify that Nedelcu e Comaneci are the authors, please delete the word.

Response 5: Yes, was done.

Point 6: Line 47 - What is the Arboblend V2 Nature? Even though you have the reference it should be briefly explained what it is. Is it a commercial product, biodegradable etc.

p.s. I see that you describe the material from line 53 to 63, please give a brief explanation anyways at this point, the reader otherwise has to comprehend it from the beginning. It will be sufficient to say "Arboblend V2 Nature, a lignin based material,"

Response 6: Yes, was added.

Lines 67-68: “the Arboblend V2 Nature material injected samples. Arboblend V2 Nature, is a lignin based material The injection moulding temperature set during”

Point 7: Line 47 - What do you mean with injected samples? Are the sample injected with the polymer or you meant extruded? Please clarify.

Response 7: It’s about injection moulding.

Lines 67-68: “the Arboblend V2 Nature material injected samples. Arboblend V2 Nature, is a lignin based material The injection moulding temperature set during”

Point 8: Line 48 - Many plastics is referred to synthetic plastics, isn't is so? Please specify.

Response 8: Yes, it’s about synthetic plastics.

Line 69: “compared to many synthetic plastics”

Point 9: Line 56 - In the introduction I would rather refer to an anonymous "producer" instead of the Tecnaro German company, which in any case should be have a more complete reference, and specify the company name, product name etc in the material an methods.

Response 9: Yes, was changed with "producer".

Point 10: Line 58-63 - Is not clear what Fiber wood is, at the beginning I thought it was a constituent of the Extrude BDP "Pearl", but reading the next sentence is cleat that is a polymer per se that you are comparing with the Arboblend V2 Nature. Thus, please rephrase explaining what Fiber wood, meaning why you are using it, why you use it as a comparison, is it the market choice for biodegradable extrusion polymers?

Response 10: We used Fiber wood because is PLA with added wood fibers and therefore has many of the same properties as PLA, the most used biodegradable material in FDM printing process.

Lines 80-83: “Fiber Wood filament is a thermoplastic material, made entirely from natural wood, which is part of its constituent elements, being produced by Fiberlogy. Fiber Wood is PLA with added wood fibers and therefore has many of the same properties as PLA, the most used biodegradable material in FDM printing process.”

Point 11: Line 71-71 - Can you please explain do you mean with wire manufacture from based material and reinforced materials?

Response 11: Was added the material: “wire manufacture by extrusion of Arboblend V2 Nature reinforced with Extrudr BDP "Pearl"” (Lines 95-95)

Point 12: Line 73 - Why you didn't reinforce also the Arbofill Fichte liquid wood? If it's just a commercial product what is the reason to study its thermal behaviour, it should be sell with a technical sheet. And in any case what would be the novelty in this?

Response 12: We didn’t reinforce Arbofill Fichte because there is no information concerning the biodegradation rate and we found that Arboblend V2 Nature has the 45% biodegradation rate during 140 days. Please see the picture below:

From our point of view, the thermal behavior is so important not only 3D printing but also for manufacture of different products under the thermal function conditions.

Lines 95-96: “The significance of this work consists in the possibility to replace parts from plastic materials take into account the negative influence of these ones over the environment.”

Point 13: Line 75 - You should explain better which synthetic plastic materials are you willing to substitute and what you mean with "parts from plastic materials" otherwise is a to broad concept.

Response 13: The following plastic materials could be replaced by Arboblend V2 Nature reinforced with Extrudr BDP Pearl, from tensile strength point of view: ABS (40MPa), Flexible (26-43MPa), HIPS (32MPa), PP(32MPa), Metal Filled (20-30MPa). The tensile strength of Arboblend V2 Nature reinforced with Extrudr BDP Pearl is around 45MPa.

https://www.simplify3d.com/support/materials-guide/properties-table/

Also according to the reference number 19:

Line 97-103: “As for the possibility of replacing synthetic plastics with the Arboblend V2 Nature material, from tensile strength point of view, [19], these one can replace PE-40 MPa, PVDF (poly vinylidene fluoride)-43 MPa and PCTFE (ethylene copolymer)-32 MPa. The Arboblend V2 Nature tensile strength mean value is 44.05 ± 0.48 MPa, the material had a brittle behavior. Regarding the friction coefficient, 0.1376 for the biodegradable material, can substitute with success the PA 12 (polyamide)-(0.32-0.38), PP (polypropylene) - 0.3, PE-HD (high density polyethylene)-0.29, ABS (Acrylonitrile butadiene styrene) - 0.5 and PVDF (poly vinylidene fluoride)-0.23.”

Point 14: Line 75-76 - "This is a research target for the authors of this article in the next period. " is a sentence that will be more appropriate at the end of the conclusion after you already showed the novelty reached by the research.

Response 14: Was delete.

Point 15: Line 80 - please add a were in front of obtained.

Response 15: Was done.

Point 16: Line 82 - What is a Noztek touch equipment? who produce it and where? Please specify.

Response 16: Was done.

Line 110: “(produced by Noztek, Great Britain).”

Point 17: LIne 84 - 80% with 20% have no meaning if written in this form, please specify if it's % in volume or in weight and in case add a respectively in order to exactly state what the % refers to.

Response 17: Was done.

Lines 107-108: “Wires of Arboblend V2 Nature, Arbofill Fichte, Arboblend V2 Nature reinforced with Extrudr BDP "Pearl" (80% with 20% in weight). obtained from extrusion of granules”

Point 18: Line 85 - Why don't you put the explanation of the equipment used and producer before the list so you do not have to repeat it? Please change.

Response 18: Was done by changing the position of equipment description.

Point 19: Line 87 - The parameters chosen are not technological, but settings parameters of the extruder, I would rather refer to them in this way or, considering are just two parameters I would just list them.

Response 19: Was delete the technological.

Line 112: ” The parameters set on the extrusion equipment in order to obtain the..”

Point 20: Line 92-94 - Please specify what are the equipment you are referring too, what they do and from which company are produced and where.

Response 20: Was done.

Line 118: “produced by Raise3D, USA”

Point 21: Line 103 - Please specify from which company are produced and where. 

Response 21: Was done.

Lines 119-120: “(NETZSCH producer, Selb, Germany).”

Point 22: Line 109 - Please specify from which company are produced and where. 

Response 22: Was done.

Line 128: “produced by FEI Company, USA”

Point 23: Line 115 - Please specify from which company are produced and where.

Response 23: Was done.

Line 134: “(NETZSCH producer, Selb, Germany)”

Point 24: Paragraph "The SEM Analysis":

You lack of giving any discussion on the figure 1d and also the other 3 figures are not deeply discussed. Is not very clear in this way what is the point in your study of having SEM micrography.

Response 24: Was added a short comment for the figure 1d.

Lines 157-158: “The diameters values of above materials were compared with the diameter value of Fiber Wood filament (figure 1d).”

Point 25: Paragraph "The EDAX Chemical Analysis":

Is quite not clear why you are doing an EDAX and the X-ray on the samples as you already know the basic chemical composition and you are not doing any process that could change it, moreover the two techniques give the same result with the X-ray being normally more precise of the EDAX. 

The only difference could be the line scan analysis that you performed with the EDAX, but being the elements presents all composed of C and O the difference is very small between reinforced and not reinforced samples. Also the technique is not precise enough to make some speculations over the position of the fibres reinforcement when the same elements are present.

Response 25: The next comment was added:

Line 313-315: “Comparing the results regarding the chemical structure of materials, used for the injected samples, and for the prototyped ones, it was concluded that, the printing process parameters do not change significantly the thermal behavior and structure of the base material and of the reinforced Arboblend V2 Nature.”

All the changes were highlighted in the paper, with gray color.

Round 2

Reviewer 1 Report

The authors have significantly improved their manuscript. Small corrections are required. It is recommended that the authors read the text carefully and make minor corrections themselves.

Author Response

Point 1: The authors have significantly improved their manuscript. Small corrections are required. It is recommended that the authors read the text carefully and make minor corrections themselves.

Response 1: The authors re-read the entire manuscript and made corrections in order to improve it. All changes are highlighted with red colour. The paper lines where the modifications were made are:

Lines 29-30: “…technology with high growth potential as well as high performance manufacturing. According..”;

Line 39: “a necessity for many industrial applications. As…”;

Line 71: “synthetic plastics temperatures. The…”;

Line 73: “raw material, in the form of granules, made it…”;

Lines 76-78: “…different plastics. Thus, Arboblend V2 Nature materials are designed to be biodegradable or resistant depending on the intended application and Arbofill Fichte materials are high-quality compounds made from renewable raw materials and plastics and both…”

Line 117: “wire diameter size…”;

Line 124: “25x4x2mm which were…”

Line 141: “between (20-200)°C with a…”

Line 165:Figure 1. The SEM wires images:…”

Line 170:” The below graphs…”

Lines 190-191:”the specific technical literature…”

Line 207:” the specific technical literature…”

Line 292:” lignin which was…”;

Line 296:”extraction procedure from…”;

Line 306:”confirmed that the diameters of the wires obtained by extrusion are…”;

Line 322:” structure of the base material Arboblend V2 Nature and of the reinforced material Arboblend V2 Nature with Extrudr BDP "Pearl…".

Reviewer 2 Report

The authors have addressed my concerns.

Author Response

Point 1: The authors have addressed my concerns.

Response 1: Many thanks for your response. We made some changes in the paper related to English and to a better understanding of it. All the changes were highlighted in the paper, with red color.

Reviewer 3 Report

Thanks to the improvements and corrections made, the manuscript from the title: "Dynamical mechanical and thermal analyses of biodegradable raw materials for additive manufacturing" is now more readable and some of the passages that were not clear enough, and created confusion in the reader, are now well explained. Cancelling, in that way, the initial doubts about the scientific interest of the research. Also the bibliography has been update and is now fine. I would recommend publication after checking the following few last points: 

Line 59: please change "but this one, it" with ", but it"

Line 71: a full stop is missing 

Line 104: Please could you use a simple it instead of "this one"

Line 104: Please could you rephrase this part? I mean I would use a conjunction between the two sentence to highlight the consequentiality of the second from the first. (i.e. ".In fact, Arboblend V2 Nature.." or ", as Arboblend V2 Nature..") 

Line 105: Is it important to specify in this introduction point the brittle nature of the Arboblend V2 Nature? If it is, you should explain why, if not just do not mention it because it could raise confusion in the reader.

Line 113: I would change the full stop for a coma

Author Response

Point 1: Line 59: please change "but this one, it" with ", but it"

Response 1: Was the modification was made on line 56, there seems to be a slight shift in the lines of the manuscript.

Point 2: Line 71: a full stop is missing

Response 2: We added a full stop at the end of the sentence, line 68.

Point 3: Line 104: Please could you use a simple it instead of "this one"

Response 3: Was changed the "this one" with “it”, line 217.

Point 4: Line 104: Please could you rephrase this part? I mean I would use a conjunction between the two sentence to highlight the consequentiality of the second from the first. (i.e. ".In fact, Arboblend V2 Nature." or ", as Arboblend V2 Nature..")

Response 4: Yes, we reformulated the paragraph: line 101: “In fact, Arboblend V2 Nature has the possibility of replacing synthetic plastics…”

Point 5: Is it important to specify in this introduction point the brittle nature of the Arboblend V2 Nature? If it is, you should explain why, if not just do not mention it because it could raise confusion in the reader.

Response 5: We deleted that comment, line 103.

Point 6: Line 113: I would change the full stop for a coma

Response 6: Yes, was added a comma instead of a full stop, line 111.

All the changes were highlighted in the paper, with yellow color.

Also, we made some small changes in the paper related to English and to a better understanding of it. All the changes were highlighted in red color.